# scGEM: Unveiling the Nested Tree-Structured Gene Co-Expressing Modules in Single Cell Transcriptome Data

**DOI:** 10.3390/cancers15174277

**Published:** 2023-08-26

**Authors:** Han Zhang, Xinghua Lu, Binfeng Lu, Lujia Chen

**Affiliations:** 1Department of Biomedical Informatics, University of Pittsburgh, Pittsburgh, PA 15206, USA; haz96@pitt.edu (H.Z.);; 2UPMC Hillman Cancer Center, Pittsburgh, PA 15232, USA; 3Center for Discovery and Innovation, Hackensack Meridian Health, Nutley, NJ 07110, USA

**Keywords:** single cell transcriptome, topic model, gene co-expressing module, nested tree structure, cellular program

## Abstract

**Simple Summary:**

Single-cell RNA sequencing has significantly contributed to the discovery of heterogenous cellular programs that are highly expressed by distinct cell subtypes. Yet, the quantification of cell subtype-specific and shared gene co-expressing modules (GEMs) associated with cell differentiation is a challenge. Herein, we have developed scGEM to uncover such hidden GEMs and conducted a systematic evaluation of model performance as well as a comparison with existing methods. We demonstrate that scGEM has the potential to generate a better biological explanation of GEMs using simulated and real-world datasets. The positive impacts of this study can illuminate the interpretation of gene modules in single-cell transcriptome analysis and shed light on the cell–cell communication and regulatory network.

**Abstract:**

Background: Single-cell transcriptome analysis has fundamentally changed biological research by allowing higher-resolution computational analysis of individual cells and subsets of cell types. However, few methods have met the need to recognize and quantify the underlying cellular programs that determine the specialization and differentiation of the cell types. Methods: In this study, we present scGEM, a nested tree-structured nonparametric Bayesian model, to reveal the gene co-expression modules (GEMs) reflecting transcriptome processes in single cells. Results: We show that scGEM can discover shared and specialized transcriptome signals across different cell types using peripheral blood mononuclear single cells and early brain development single cells. scGEM outperformed other methods in perplexity and topic coherence (*p* < 0.001) on our simulation data. Larger datasets, deeper trees and pre-trained models are shown to be positively associated with better scGEM performance. The GEMs obtained from triple-negative breast cancer single cells exhibited better correlations with lymphocyte infiltration (*p* = 0.009) and the cell cycle (*p* < 0.001) than other methods in additional validation on the bulk RNAseq dataset. Conclusions: Altogether, we demonstrate that scGEM can be used to model the hidden cellular functions of single cells, thereby unveiling the specialization and generalization of transcriptomic programs across different types of cells.

## 1. Introduction

The advent of high-throughput single-cell RNA sequencing (scRNAseq) has enabled and revolutionized massively quantitative research in molecular biology [1,2]. Compared with previous gene expression analyses in microarray and bulk-RNAseq, the major advantage of scRNAseq is its availability to capture cellular heterogeneity and diversity in different cell subtypes [3,4,5]. Cell subtype clustering methods based on principle component analysis (PCA) and the Louvain method of community detection have been widely used and shown to be applicable for subtyping [6,7,8]. While clustering analysis reveals a categorical subpopulation of cells sharing characteristics at the whole transcriptome level, it fails to quantify and represent the various cellular programs that each cell would employ in differentiation and specialization [9,10]. Essentially, the transcriptome of a single cell consists of modules of genes with expressions that are regulated by cellular transcriptomic processes [11,12,13]. Differences in these cellular transcriptomic processes underlie the heterogeneity within subpopulations of cells. Thus, discovering such processes can lay a foundation to gain further insights into cellular transcriptomic programs and the processes by which cells are regulated, e.g., through cell-intrinsic differentiation programs or cell–cell communication [14,15].

Recently, topic model methods are being used in the context of scRNAseq to extract distributions of hidden gene co-expressing modules (GEMs) over cells [16]. Non-negative matrix factorization (NMF) is used to search the cellular function signals that explain the heterogenous programs among single cells [17,18,19,20]. Latent Dirichlet allocation (LDA) and hierarchical Dirichlet processes (HDP) have been utilized to find cell subtypes and links between mutations and gene expression [21,22,23,24,25,26]. Compared with PCA, topic models can provide a better explanation of the top genes inside the GEMs and components [27,28]. However, these algorithms share common limitations in that they assume the generative processes in a cell to be independent [29] and restricted to cell subtypes, which violates the knowledge that cell differentiation (specialization) follows a major lineage program that orchestrates mixed transcriptome functions in the cell, resulting in upregulation of varied pathways. For example, CD8^+^ T cells can acquire regulatory functions [30,31], and high proliferative and regulatory signals in CXCL13^+^ T cells have been reported [32,33]. A new computational framework that relaxes this assumption is needed. There are other methods developed to detect GEMs in single-cell transcriptome data, such as scWGCNA [34], which extends the use of WGCNA [35] to the single-cell context. Nevertheless, these methods only return the average expression of GEMs in each cell and membership of genes within GEMs. The relative proportion of gene modules that are recruited by each cell remains undefined. 

In order to model the transcriptomic programs of single cells more accurately, we introduce scGEM, a nested tree-structured generative model that identifies subtype-specific and shared GEMs via scRNAseq data (Figure 1). scGEM is based on the nested hierarchical Dirichlet processes (nHDP) model [36] and explicitly represents the hierarchical relationships of the transcriptomic processes using a customized tree structure. As a nonparametric Bayesian model, it allows for transfer learning from the pretrained distribution of GEMs in prior analyses. Using simulated scRNAseq, we show that scGEM outperforms other topic model methods in terms of predictive likelihood and biological explanations of GEMs, both of which can be further improved by seeing a larger number of cells and using a pretrained model as a prior. In addition, we demonstrate that scGEM can discover both cell subtype-specific GEMs and GEMs shared by cells across the boundaries of conventional cell subtyping in public datasets of peripheral blood mononuclear cells (PBMC) and early human brain development scRNAseq data [37]. Finally, we built an scGEM model using triple negative breast cancer (TNBC) scRNAseq data and deconvoluted the GEM signals with experimentally validated results in an additional TNBC bulk-RNAseq dataset. Overall, we show that scGEM is a useful tool to decipher credible cell type-specific and shared GEMs in a tree-structured way.

## 2. Materials and Methods

### 2.1. Data Collection and Preprocessing

Publicly available PBMC 3K, 8K, and 33K scRNAseq datasets were acquired from 10× Genomics Support. (https://www.10xgenomics.com/support/single-cell-gene-expression (accessed on 16 July 2023)). In the PBMC datasets, single cells with (1) less than 200 genes, (2) over 2500 genes, or (3) over 5% mitochondrial genes were removed during quality control. The downstream cell subtyping analysis of PBMC 3K followed the Seurat tutorial (https://satijalab.org/seurat/articles/pbmc3k_tutorial.html (accessed on 16 July 2023)). The processed scRNAseq and fully annotated metadata on early human brain development [37] were obtained from UCSC Cell Browser (https://cells-test.gi.ucsc.edu/?ds=early-brain (accessed on 16 July 2023)). The scRNAseq of TNBC in Zhang’s TNBC study was downloaded with accession ID GSE169246 [32] (https://www.ncbi.nlm.nih.gov/geo/ (accessed on 16 July 2023)). Single cells sampled from baseline tumors were used in this study for further clustering and annotation. The unique molecular identifier (UMI) counts of each cell were log-normalized and then scaled over all cells. The percentage of mitochondrial genes was regressed out during scaling process. The above steps were performed in Seurat v4 [38] with the default settings. The first fifteen principal components were used to find neighbors. We used a resolution of 0.8 for the Louvain clustering. The cell types were then annotated with canonical marker genes: *CD3D*, *CD8A*, *FOXP3*, *IL7R*, *NKG7*, *GNLY* for T and natural killer (NK) cells, *CD79A*, *MS4A1*, *MZB1* for B and plasma cells, and *CD14*, *CD68*, *CD1C*, *S100A8*, *C1QA*, *LYZ* for myeloid cells. The bulk-RNAseq data (*n* = 342) and associated clinical information of WGS560 breast cancer [39] were downloaded from the supplementary tables of the original study (https://pubmed.ncbi.nlm.nih.gov/27135926/ (accessed on 16 July 2023)). We extracted 73 TNBC samples for downstream analysis.

### 2.2. Tree-Structured Gene Co-Expressing Modules

As a nonparametric Bayesian model based on a nested hierarchical Dirichlet process, the initiation of scGEM involves drawing the base distributions from a Dirichlet prior (DP) in a hierarchical way such that $G_{d}|G$ is independent and identically distributed as $\mathrm{DP}\left(\tau G\right)$, where $G∼\mathrm{DP}\left(\alpha G_{0}\right)$ (as suggested in previous works) [36,40]. An applicable construction of above is through the stick-breaking process [41] below:$$G_{d}=\sum_{i=1}^{\infty }\pi_{i}^{d}\prod_{j}^{i-1}\left(1-\pi_{j}^{d}\right)\delta_{\phi_{i}}$$
where $\pi_{i}^{d}∼\mathrm{Beta}\left(1,\tau \right)$ and $\phi_{i}∼G$. $\pi_{i}$ can be considered as the relative proportion that is broken from rest of stick, of which the total length is 1, and $\delta_{\phi_{i}}$ is an indicator function that determines whether atom $\phi_{i}$ is realized. Another practical construction method for hierarchical DP is the Chinese restaurant process [42], shown below.
$$\phi_{n+1}|\phi_{1},\ldots ,\phi_{n}∼\frac{\alpha }{\alpha +n}G_{0}+\sum_{i=1}^{n}\frac{1}{\alpha +n}\delta_{\phi_{i}}$$

To faster initialize the prior GEM tree, we first normalize the UMI count of each gene within a certain cell to relative proportions. Using a preset training size, we first used L1 KMeans to find the centroids of each node on the top level of the tree, then ‘breaking’ the relative proportions of each gene and renormalizing. We repeated this process iteratively on the other level of the GEM tree until reaching convergence. To begin the tree initialization, it is necessary to preset the tree structure and provide the parent–child relationships on each level. For example, a tree structure of 5-4-3 means five GEMs on the top level, four second-level child GEMs under each parent GEM on the top level, and three third-level child GEMs of each parent GEM on the second level. Ideally, the number of top-level GEMs should be larger than the number of major cell types within the scRNAseq data in order to prevent underestimation. When delving deeper into the GEM levels, the interpretation of GEM tends to become more distinct and less generalized. The signals of cellular programs are preserved on each level during the stick-breaking process in the “break the information” step below (Algorithm 1). Hence, while the top-level GEMs are suitable for capturing broad signals, the GEMs at the leaf nodes correspond to highly specialized functions.
**Algorithm 1:** scGEM Tree initialization(1)For each cell *d*, normalize the gene count(2)For each tree level *l*
For each GEM *g*
Estimate centroids Θ with L1 norm and cluster assignments *S*For each cell *d* assigned to *s*
Break the information’ by Θ_s_Renormalize as in (1)

We used the L1 norm, as it can provide more reliable and less sensitive clustering performance in sparse single-cell data [43,44]. The variational Bayesian method [45,46] was used to infer the posterior, and a switching parameter γ was set in the search step to encourage the node to traverse down to the leaf node. A lower switching parameter results in a conservative search strategy that saves more sufficient statistics on higher levels of the tree. The relevant R package scGEM/v0.1.2 is provided on GitHub (https://github.com/hansolo-bioinfo/scGEM (accessed on 24 August 2023)).

### 2.3. Model Performance Evaluation

#### 2.3.1. Simulation

To simulate single-cell transcriptome data, we used R package splatter/v1.22.1 to imitate the distributions of the UMI count within each cell [47]. There were 11,000 single cells, from which ~14,000 genes were generated using the log-normal distribution of the UMI counts. To split the dataset into training and testing sets for model evaluation on different training sizes, we first sampled 1000 cells as testing set, then randomly selected another 1000 cells from the remaining 10,000 cells as a training set. We gradually increased the size of the training set from 1000 to 10,000 in steps of 1000. This procedure helped to organize ten training sets of different sizes. We replicated the above procedure three times in order to remove any selection bias.

#### 2.3.2. Implementation of NMF and LDA

We used the R packages RcppML/v0.3.7 [48] and topicmodels/v0.2.14 [49] to train the NMF and LDA models for the sparse UMI count matrix. The topics were set to the corresponding total number of GEMs in the scGEM. The distributions of terms over topics in LDA were used to infer the distributions of topics over cells in LDA. As for inference in NMF, as V≃WH, where **W** denotes the distributions of genes over GEMs and serve as the initial matrix in factorizing the testing data. **H** denotes the distribution of GEMs over test cells. **V** denotes the expression matrix of testing data.

#### 2.3.3. Evaluation Metrics

Perplexity was used to measure the performance of the scGEM probabilistic model [50]. It is often considered as a validate metric for model selection in topic models, as it evaluates the predictive likelihood per token in the testing set. Lower perplexity indicates better performance of the model. After the learning process converged in the training set $\beta_{k,g}$, the distributions of genes over *K* GEMs were used to infer GEM distributions $\theta_{d,k}$ in the testing data. The predictive likelihood of the testing set was then estimated as:$$L\left(\mathrm{test}\right)=\sum_{d=1}^{D}\sum_{g=1}^{G}C_{d,g}\cdot \log\left[\sum_{k=1}^{K}\theta_{d,k}\beta_{k,g}\right],$$
where $C_{d,g}$ refers to the UMI count of gene *g* in testing cell *d*. The perplexity of the total testing cells was then estimated as
$$\mathrm{Perplexity}\left(\mathrm{test}\right)=\exp\left\{-\frac{L\left(\mathrm{test}\right)}{\sum_{d=1}^{D}\sum_{g=1}^{G}C_{d,g}}\right\}$$

We applied a topic coherence metric to evaluate the coherence of topic genes in the learned GEMs. The coherence of each GEM was estimated using the sum of all pointwise mutual information (PMI) of any gene co-expressing pair $p\left(g_{i},g_{j}\right)$ within the top 50 genes of a GEM. A higher topic coherence score usually indicates better consistency of the marker genes, and is more likely to match the experience of a human expert [51].
$$\mathrm{PMI}\left(g_{i},g_{j}\right)=\log\frac{p\left(g_{i},g_{j}\right)}{p\left(g_{i}\right)p\left(g_{j}\right)}$$

### 2.4. Gene Expression Analysis

Cibersort [52] was used on the gene expression data of WGS560 to estimate the 22 immune cell absolute signals and their relative proportions in the tumor microenvironment. The T cell-inflamed and cell cycle gene expression profiles (GEP) signatures [53] for all TNBC samples in WGS560 were assessed by rank-based statistics for the inflamed and cell cycle canonical gene sets in the original paper. To estimate the cell functions from the scGEM results, we used gene set variation analysis (GSVA) [54] for the top 50 weighted genes in each GEM. The GSVA scores for all associated samples in a certain GEM were than scaled to 1.

### 2.5. Cosine Similarity among GEMs and Marker Genes

To measure the correlations between GEMs and highly variable marker genes in each cell cluster, we first computed the differentially expressed genes in each cell subtype and selected significant genes (adjusted *p*-value < 0.001 and average log_2_ fold change > 0) using the default parameters in the *FindAllMarkers* function in Seurat. We then calculated the cosine similarity between the marker genes in each cell subtype and the top 150 genes in each GEM. Cosine similarity was used in the comparison between scGEM and modules based on scWGCNA, NMF, and LDA. To generate scWGCNA results, we first calculated the pseudocells from Seurat objects and then ran the scWGCNA process using the default settings. Assuming a and b to be two vectors of words, the cosine similarity was then computed as shown below. A higher cosine similarity suggests a better correlation between the meaning of the two GEMs or the explanation of a GEM and a certain cell trait.
$$\cos\mathrm{ine}\left(\theta \right)=\frac{a\cdot b}{\left.|a\right.|\left.|b\right.|}$$

### 2.6. Statistical Analysis

One-sided Wilcoxon rank sum statistical tests [55] were used to compare the topic coherence between model and learning strategies, as the number of topics on each level is dependent on the tree construction; the contents within each topic on the same level might be altered due to the random initialization in the mini-batch learning. Paired Wilcoxon rank sum tests were specifically used to test differences in perplexity, because we fixed the training samples for different models. Therefore, the tree construction is the only factor that affects the perplexity. Kruskal–Wallis statistical tests [56] were utilized to test the difference of variance in T cell CD8 signals on multiple lymphocyte infiltration levels. Missing data (n = 18) were removed during all Kruskal–Wallis statistical tests.

## 3. Results

### 3.1. scGEM Identifies High-Resolution Gene Co-Expressing Modules in the PBMC Dataset

To investigate how scGEM enables GEM detection in scRNAseq, we first processed the well-annotated PBMC 3K dataset, then constructed a 5-4-3 three level scGEM tree structure (Figure 2a) to better capture the underlying cellular programs from the CD45^+^ cells (T cells, B cells, NK cells, Myeloid cells, and a small number of other cell types within the dataset). Five general GEMs were initialized on the top level of the searching tree; four child GEMs were subsequently created under each parent node on the first level, and three specialized child GEMs under their parent node on the second level were finally built as the leaf node. After training converged, we summarized the single cells that expressed a certain GEM and used a fan tree to represent the distribution of cell types in each GEM (Figure 2a). 

In total, 85 GEMs were initialized at the starting point. We found that functions of myeloid cells could be fully captured by GEM 2 and its descendants (Figure 2b,c). As shown, GEM 2 covers most of the myeloid-related cells in the UMAP (monocytes, dendritic cells, and platelets). The top three genes of GEM 2 are *LYZ*, *S100A9*, and *S100A4*, which are often used individually or together as gene markers for cell subtyping in the myeloid lineage [57,58], suggesting a common program of this lineage. More interestingly, the specialized GEMs on the third level demonstrate the capability to detect shared and specific programs in myeloid cells. For example, GEM 38 mainly represents the specific function within platelet cells, as its markers are common in subtyping (*PPBP*, *PF4*, *GNG11*) [59]. GEM 39 (*S100A8*, *TYROBP*, *LGALS1*) and GEM 47 (*PLBD1*, *IL8*, *QPCT*) exhibit distinct functions in CD14^+^ monocytes. GEM 39 is shared between CD14^+^ monocytes and Naïve CD4 T cells, as it reveals the characteristics of modulating inflammation, whereas GEM 47 is more unique to monocytes in that it reveals an antimicrobial function in classical monocytes [60]. GEM 42 (*FCER1A, CLEC10A, HLA-DQA/B*) denotes strong signals of type 1 conventional dendritic cells [61]. GEM 45 (*FCGR3A, MS4A7, CKB*) represents mediation functions such as cytokine production in response to immune complexes in non-classical CD16^+^ monocytes [62]. The top three genes within GEM 45 are also the marker gene in selecting FCGR3A^+^ monocytes from PBMC 3K cell clusters. We show the distributions of cell types in each myeloid GEM in Figure 2c. We found that GEM 42 is more expressed in dendritic cells than monocytes, albeit it is a shared transcriptomic process in monocytes as well, while GEM 45 is intuitively more supported by monocytes. These findings are consistent with cell subtyping gene markers for annotating dendritic cells and FCGR3A^+^ monocytes [63]. Altogether, we showed that scGEM recognizes the common characteristics and subtle differences in expression in myeloid cells, suggesting that different types of myeloid cells employ similar cellular functions, though on different levels [64].

In CD8/NK-related cell clusters, GEM 4 (*NKG7, CCL5, GNLY*) catches the common signals of natural killing and cytotoxic process (Figure 2d,e). We noticed that CD8 T cells consist of multiple specific and shared programs with NK cells at high resolution that cannot be observed with other methods (Appendix A and Figure 3e) or previous marker-based approach (Figure 3a). For example, GEM 63 (*GZMB, CCL3, FGFBP2*) adds more cytotoxic markers on top of GEM 4, and shows CD8 T cell activation [65]. GEM 65 (*CCL4, IGFBP7, TYROBP*) may imply regulatory function via insulin-like growth factor [66], and CCL4 acts as the production of activated NK cells. GEM 68 (*GZMK*, *TIGIT*, *LAG3*) indicates a transitionally dysfunctional program [67], whereas GEM 69 (*GZMH*, *SAMD3*, *IL32*) implies a proinflammatory and effector state in CD8 cells [68]. The shared program GEM 71 (*HOPX*, *SELL*, *XCL1*) suggests a common function of early development of differentiation pathways in both CD8 and NK cells [69]. The stacked violin plot reveals that CD8/NK-related cells can be quantified as mixture distributions of GEMs (Figure 2e). As is illustrated in Figure 2a, the GEM tree can be separated into four major parts (CD4 GEMs, CD8/NK GEMs, B GEMs, and Myeloid GEMs) with respect to the common programs on the first level and specific functions on the leaf node. The number of colors in each pie chart indicates the degree of generalization of this GEM. Apparently, CD4 and myeloid GEMs (e.g., GEM 5 and 12) share more functions with other cell types, suggesting their importance in connecting, assisting, and regulating the cellular signaling network [70,71]. Altogether, scGEM provides a higher-resolution and parent–child perspective for measuring single cell functional programs.

### 3.2. Comparison of scGEM and Marker Genes in Cell Type

To further explore the meaning of gene coexpressing modules, we calculated the differentially expressed genes in each cell cluster and used the cosine similarity metric to measure the correlation between GEMs and those highly variable genes. Interestingly, we found that each cell type can be represented by a set of GEMs in the same lineage (Figure 3a), which is consistent with our previous analysis. In Figure 3a, the GEMs are separated into five blocks with respect to their associated lineage and parent nodes. For example, GEM 4 and its child nodes apparently exhibit strong cosine similarity with the differentially expressed gene makers in the CD8 T and NK clusters. We then looked into the GEM–GEM correlations to determine whether the GEMs are independent. We depicted the pairwise cosine similarity among GEMs in the Myeloid GEM block (Figure 3b) and TNK GEM block (Figure 3c), finding that the average GEM–GEM correlation significantly decreased when moving down the tree level. In Myeloid GEMs (Figure 3b), the averaged correlation between GEMs was significantly higher at level 1 and level 2 (GEM 2, 10, 11, 12, 13) than at level 3 (*p* = 7.981 × 10^−6^). Again, the averaged correlation between the top-level TNK GEMs was significantly higher than at level 3 (*p* = 0.0003), implying that the GEMs become more specialized and less common at the lower levels of the tree (Appendix A). In addition to the highly variable marker genes within each cell subtype, we employed scWGCNA to identify 20 modules and compared these with the modules found by scGEM (Figure 3d). However, these failed to reach the fine resolution that scGEM provides. A limited number of modules were highly associated with cell type-specific marker genes (Appendix A). Consequently, the mixed distribution of GEMs in the same lineage can represent scWGCNA modules as well (Figure 3e).

To validate our conclusion, we analyzed the scRNAseq of early human brain development. The scGEM tree was constructed using the same 5-4-3 structure in the PBMC 3K dataset, as there are four major areas (central cortex, frontal cortex, occipital cortex and telencephalon) in 45,156 cells. Likewise, we found that the fully annotated area and Carnegie stage can be represented by different GEMs in different lineages, suggesting that GEMs have the potential to comprehend cellular programs for such differentiation processes (Appendix A). There were 25 modules detected by scWGCNA (Appendix A). Nonetheless, each area and Carnegie stage can be captured by only few modules (Appendix A), consistent with our conclusion in PBMC 3K that scGEM provides higher resolution in identifying and quantifying the cellular programs.

### 3.3. Systematic Review of scGEM Model Performance

Because the initialization of the learning process is critical to model performance, we utilized simulation data and larger datasets to evaluate scGEM under different initialized tree structures and implementations. We first used PBMC 33K, a dataset with the same donor as PBMC 3K, to examine the effect of the GEM tree size and number of training samples on the predictive perplexity. A number of 3000 cells were held unseen as testing samples. We then randomly selected a number of training samples, and created GEM trees in 5-4-3, 4-3-2, and 3-2-2 structures, which resulted in 85, 40 and 21 total GEMs in each starting global tree, respectively (Figure 4a). As described, PBMC datasets mainly consist of immune cells. Therefore, the 5-4-3 structure needs to be slightly larger in order to represent the number of major cell types in the data, while 3-2-2 is smaller in the case of a smaller number of cell types within due to the random selection of the training cells. 

The learned gene distributions over the GEMs in the training model were then used to infer UMI count distributions in the testing cells. We employed predictive perplexity on the testing cells to measure the performance of scGEM in terms of predicting likelihood per UMI count. The whole testing process was conducted three times to diminish the impact of random initialization. As expected, the perplexity of the testing cells decreased as the number of training cells increased [36]. We noticed that the size of the GEM tree has a significant effect on model performance. The *p*-value between perplexities for a small GEM tree (K = 21) and a larger tree (K = 85) is significant (*p* = 0.001), indicating that the information of cells fails to be generated as well when using the limited GEM tree. Additionally, we found that scGEM using a larger tree outperforms the medium tree (K = 40) when the number of training cells is over 18K (*p* = 0.031). Overall, scGEM using a large initialization tree shows better performance as the number of training cells increases. In order to take advantage of previously obtained scRNAseq knowledge, we pretrained the starting parameters in PBMC 8K and then transferred them into the scGEM (K = 85) model for PBMC 3K. We selected PBMC 8K as the pretraining dataset, as the donor is different from the one in PBMC 3K. Therefore, PBMC 8K is an excellent candidate to simulate real-world studies in which the model would be pretrained using different tumors from the same cancer type. As demonstrated in Figure 4b,c, scGEM shows a higher total likelihood, faster convergence speed, and increased topic coherence after taking advantage of the knowledge gained from previous scRNAseq data; the *p*-values of the topic coherence between pretrained scGEM and mini-batch scGEM as well as between mini-batch scGEM and one-batch scGEM are 0.001 and 0.091 respectively). 

To comprehensively review the scGEM model, we tested the perplexity and topic coherence on the simulation dataset and compared the performance with other existing topic models. Similar to the data separation in PBMC 33K, we split the simulation data into testing (n = 1000) and a range of sizes of training sets (n = 1000 to n = 10,000). NMF, LDA, scGEM with conservative parameters (denoted as scGEM shallow), and scGEM models were constructed. The difference between scGEM shallow and scGEM is that scGEM shallow does not encourage the search process to traverse down to the leaf node, i.e., the switching probability is set to 0.5. As depicted in Figure 4d, the predictive likelihood per UMI count (perplexity) is always lowest for scGEM in all sizes of training samples and initialization tree sizes. Moreover, scGEM shallow lies between scGEM and LDA, which can be ascribed to its more conservative parameters. LDA models exhibited better performance in perplexity and were worse in topic coherence compared with NMF-based methods (Figure 4e). More interestingly, we found that topic coherence on the second and third level of scGEM were significantly higher than NMF (*p* = 0.4613 in K = 21; *p* < 0.001 in other comparisons). As the size of the GEM tree becomes larger, the topic coherence between scGEM and scGEM shallow become more significant, suggesting that the conservative search parameters impede biological explanation of the top-weighted genes in each GEM.

### 3.4. scGEM Deconvolutes Function Signals in Bulk Transcriptome Data

As we have unveiled the hidden GEMs using single cell transcriptome data, it is of interest to apply such knowledge back to discovering functional signals in tumors with bulk transcriptome data. We examined the utility of scGEM from this perspective by applying the model to untreated TNBC single cells (n = 89,219) to construct a 5-4-3 GEM tree. We annotated ten cell subtypes within the baseline TNBC single cells (Figure 5a). The GEMs exhibited in at least 1% of the cells are shown in Figure 5b. As expected, the GEMs on the first level of the tree each successfully capture the general signals of a major cell type. GEM 1 is expressed by almost all types of T cells, especially CD4 and T regulatory cells. GEM 2 and GEM 4 represent the signals of B and plasma cells, respectively. GEM 3 reveals a combination of both CD8 and NK cells, which shows crosstalk between them. GEM 5 indicates a common program of myeloid cells shared by monocytes, dendritic cells, and macrophages.

While the GEMs on the first level display the universal functions in the major cell types, we found that the GEMs on the second level show shared but constraint programs among a few cell subtypes. For example, GEM 9 (*TIGIT*, *CTLA4*, *FOXP3*) is highly expressed in regulatory and proliferative T cells, suggesting the co-existence of these two T cell subtypes under the parent GEM 1 (*CXCR4*, *S100A4*, *IL7R*) (Figure 5c). However, GEM 14 shows co-expression of CXCL13^+^ CD8 as well as proliferative T cells (Figure 5d), which indicates a potential relationship between T proliferation and CD8 and regulatory T cells [32,72,73,74]. Compared with the search path of GEM 14, GEM 9 relates GEM 2 (B cell common programs) and leaf node GEM 36 (CXCL13^+^ CD4), which is consistent with previous studies showing interaction between B cells and T follicular helper cells [75]. Conversely, GEM 14 is correlated with GEM 17 (NK common programs), implying a co-regulatory relationship between these two programs [76]. 

Finally, we compared the function signals using the top genes in the GEMs to infer the CD8 lymphocyte infiltration and cell cycle signatures in an additional WGS560 breast cancer dataset. We used the GSVA scores of the top 50 genes in GEM 3 (CD8/NK common program) to assess the T cell CD8^+^ signals and used GEM 37 and 52 to assess cell cycle signatures, as they are highly expressed in proliferative cells. To compare our estimation with other methods, we performed a calculation for Cibersort with absolute value, somatic copy number alteration (SCNA) [77], and inflamed GEP signatures using the same expression data (Figure 6a,b). The results show that the Cibersort method fails to correlate the estimated CD8 signals with lymphocyte infiltration (*p* = 0.34). GEP signatures exhibit moderate significance among different infiltration levels, though the variance is limited (*p* = 0.02). Predicted CD8 signals by scGEM demonstrate significant differences in variance with respect to infiltration (*p* = 0.009). Interestingly, SCNA fails to show a correlation with lymphocyte infiltration (*p* = 0.124). As for cell cycle signatures, both GEM scores (*p* < 0.001) and GEP signatures (*p* = 0.002) exhibit more significant *p*-values than SCNA (*p* = 0.023, Spearman = 0.31). Overall, these results demonstrate that GEMs are better able to reflect the state of the transcriptional process compared to other state-of-the-art measures. Moreover, they show that the intricate relationship between GEMs can greatly enhance the analysis of regulatory networks and enable the deconvolution of signaling pathways in bulk RNAseq data.

## 4. Discussion

scRNAseq has transformed molecular research, yet the advantages of using previously generated data and the identification of hidden GEMs in sparse gene expression matrices remain unmet needs. To resolve this problem, in this paper we propose scGEM, a nested nonparametric Bayesian tree-structure model that detects GEMs via scRNAseq. We demonstrate that scGEM can successfully recognize gene modules at a high resolution even in PBMC 3K, which has a limited sample size. We show that the pretrained scGEM in a larger PBMC 8K dataset (with a different donor) resulted in improved the model performance in terms of convergence, total likelihood, and the coherence of the most weighted marker genes in each GEM. Therefore, the GEMs learnt from scGEM are transferrable, reflecting good generalization performance across different scRNAseq datasets. Using simulated single-cell and PBMC 33K datasets, we show that scGEM with deep searching outperforms other similar GEM extraction methods such as NMF and LDA. In the end, our validation using real-world TNBC data shows that scGEM can deconvolute the signals of CD8 T cells more accurately than Cibersort or the SCNA score, which is due to the fact that GEMs have better ability to capture functional signals and take into consideration the crosstalk among subtypes.

The parameter tuning of scGEM was decided according to lower perplexity on the testing dataset and higher topic coherence, as suggested in previous evaluations of topic models [78,79]. In practice, we recommend that in the context of scRNAseq, the expected size of the GEM tree should match the biological essence. For example, if the input single cells are sampled from one major cell type (T cells, B cells, etc.), a smaller tree fits better with the known number of subtypes of these cells and avoids overestimation of the number of GEMs inside the cells. On the contrary, if the input single cells are samples from multiple types (i.e., CD45^+^ or tumor microenvironment), a larger tree is more appropriate in order to ensure that scGEM does not underestimate the number of GEMs. A conservative searching parameter results in underestimating the complexity of GEMs when single cells are well differentiated. Therefore, an understanding of the relevant biological questions is essential to the use of this topic model. 

One limitation of this work is the removal of ‘nonsignificant’ genes prior to GEM tree initialization. While a few scRNAseq studies have deleted mitochondrial genes, heat shock protein genes, ribosomal protein genes, or housekeeping genes before using the PCA to remove noise signals, the removal of genes in scGEM needs to be carefully considered. More detailed research on housekeeping genes and their co-expression relationships with other genes is required. Another limitation lies in the hidden assumption that the distribution of gene UMI counts in different batches are assumed to be independent and identically distributed; otherwise a pretrained model from a larger dataset will simply fall. Therefore, comprehensive analysis of batch effects before transfer learning would help to refine learning performance. Interestingly, a recent review has shown that topic models can generate robust and dense modules with low noise that are able to identify the differences in regulatory relationships [80]. Future work is needed to explore the assessment of cell–cell communication and regulatory networks using the interaction score between GEMs and their underlying biological explanations. 

## 5. Conclusions

The comprehensive and effective quantification of the underlying GEMs that direct cell differentiation and specialization can help to pave the way for personalized medicine. In this study, we introduce scGEM, a tree-based nonparametric Bayesian topic model, to identify both cell type-specific and shared GEMs in single-cell transcriptome data. We first evaluated the meaning of GEMs in PBMC and neuron single-cell datasets, then validated the performance of scGEM using simulation datasets. A larger sample size of training cells, a sophisticated tree structure that matches biological essence, and prior knowledge that takes advantages of previous single-cell analyses are shown to significantly improve model likelihood. scGEM exhibits lower perplexity and higher coherence of the top genes than existing NMF and LDA-based methods in unseen single cells. Additionally, we studied GEMs learned from TNBC and associated the GEM scores using top genes in an additional TNBC validation dataset. Enhanced correlations between GEM scores and lymphocyte infiltration/cell cycle were observed compared to state-of-the-art scores. Altogether, our findings in this study illustrate the use of scGEM to unveil and quantify hidden GEMs that lead to cell heterogeneity. We envision that the performance of scGEM will be massively improved as it “sees’’ more single cells in the future. 

## Figures and Tables

**Figure 1 cancers-15-04277-f001:**
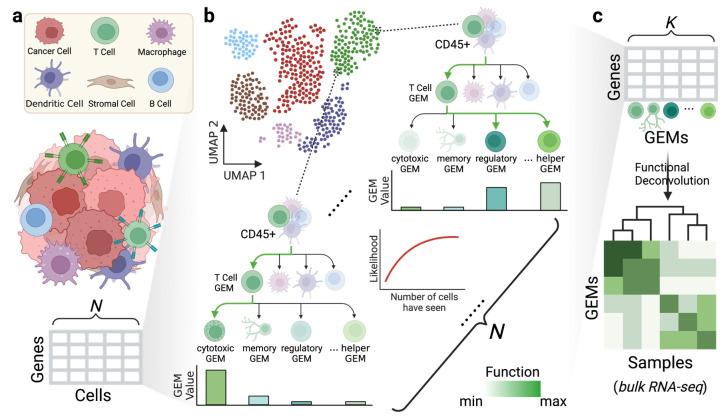
Overview of scGEM. (**a**) An illustration of the tumor microenvironment is shown. The count matrix below represents the format of expression in single-cell analysis, where each row refers to the gene name and each column to the cell index. (**b**) scGEM uses the count matrix as the input to generate nested tree-structured GEMs. In the tree structure, the color of the function GEM becomes darker and more solid as the function signal increases, while the color of the function GEM becomes lighter and transparent as the function signal decreases. The path of functions used in certain cell is labeled in solid green. (**c**) After the learning process achieves convergence, scGEM returns the distribution of GEMs and an associated gene weight matrix, in which each column represents a GEM and each row represents a gene. The top-weighted genes are then used in the functional deconvolution of the bulk-RNAseq data.

**Figure 2 cancers-15-04277-f002:**
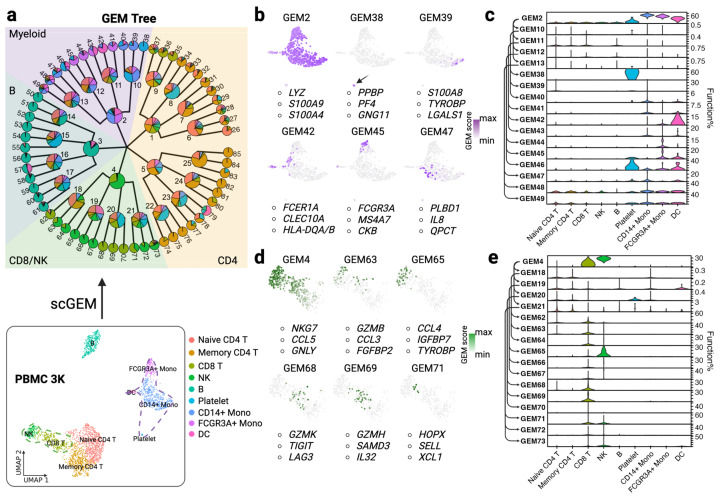
scGEM-identified GEMs across cell types in the PBMC 3K dataset. (**a**) The GEM tree of the PBMC 3K dataset is shown. Each GEM is illustrated in a pie chart, the color of which indicates the distribution of cell types that activate this functional program. The UMAP of the filtered PBMC 3K dataset (n = 2638) is shown below. The colors of cell subtypes in the fan plot are the same as the labels in the UMAP. (**b**,**d**) Zoomed-in view of the Myeloid (**b**) and CD8/NK (**d**) cell clusters. Selective GEMs that represent cell type-specific and shared programs are shown. The top three unique genes of each GEM are listed below the associated GEM UMAPs. The arrow in (**b**) indicates the position of GEM38. The function signals are revealed in gradient purple and green colors for the Myeloid and CD8/NK programs, respectively. (**c**,**e**) The distributions of all Myeloid GEMs (**c**) and CD8/NK GEMs across cell types are organized in stacked violin plots. The curved arrow in the Y-axis refers to the parent–child relationship between GEMs.

**Figure 3 cancers-15-04277-f003:**
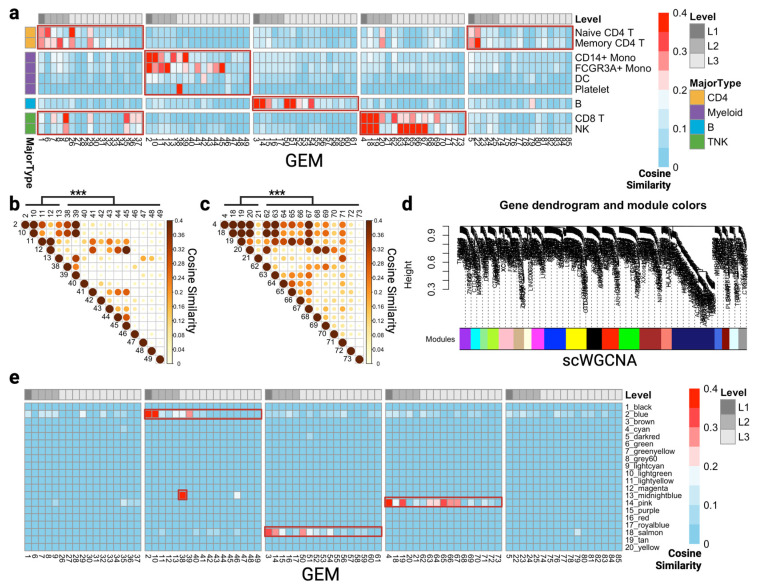
Correlations among GEMs and marker genes within cell types in the PBMC 3K dataset. (**a**) Heatmap of cosine similarities between differentially expressed genes of cell subtype in PBMC 3K and top 150 genes of total 85 GEMs by scGEM. GEMs are labeled with respect to their tree levels and cell types are labeled based on their major cell types. (**b**) GEM–GEM correlations in Myeloid GEMs measured by cosine similarity. (**c**) GEM–GEM correlations in TNK GEMs measured by cosine similarity. (**d**) Gene co-expressing modules that were detected by scWGCNA. In total, 20 modules were found. (**e**) Heatmap of cosine similarities between module genes by scWGCNA and top 150 genes of 85 total GEMs by scGEM. GEMs are labeled based on corresponding tree levels. The Wilcoxon rank sum test was used to test the difference in cosine similarities between tree levels in (**b**,**c**). *** denotes *p*-values less than 0.001. The red frames in (**a,e**) indicates GEM lineage that represent the associated traits.

**Figure 4 cancers-15-04277-f004:**
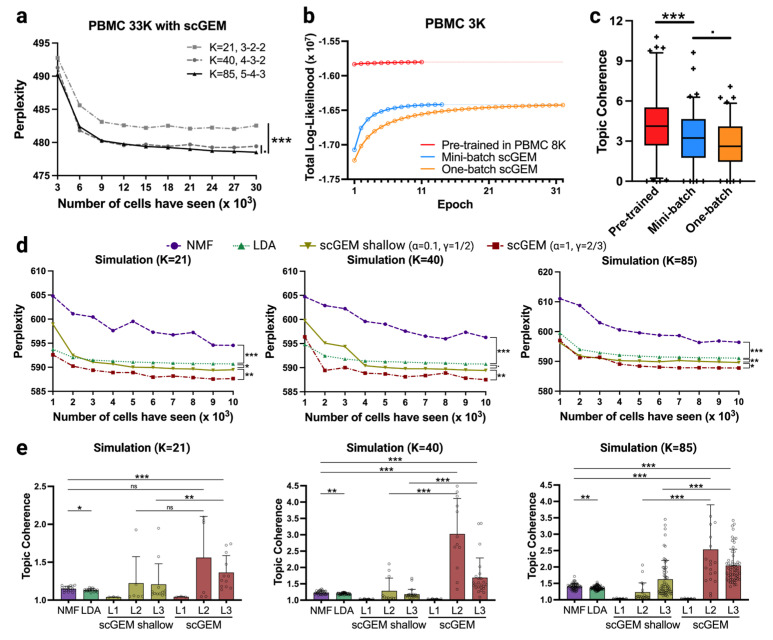
A systematic review of scGEM performance in simulation data. (**a**) The averaged perplexities using different numbers of training samples on PBMC 33K are shown. The X-axis represents the number of training samples, starting from 3000 and increasing to 30,000. The Y-axis represents the total perplexity of the testing samples (n = 3000). The curves of the solid, dashed, and dot-dashed lines show the perplexities of the tree training models (K = 85, K = 40, K = 21), respectively. (**b**) The learning curves of different learning strategies on model (K = 85) are plotted. The X-axis means the number of epochs, starting from 1, while the Y-axis means the corresponding total likelihood after each epoch. The one-batch algorithm converged at epoch 31. Mini-batches with a learning rate of 0.01 converged at epoch 14. The pretrained model followed by a mini-batch with a learning rate of 0.01 converged at epoch 11. Curves are colored as respectively depicted in legend. (**c**) The distribution of the topic coherence of model (K = 85) are illustrated in 5–95% boxplots, with outliers labeled as “+”. (**d**) The average perplexities using different numbers of training samples in the simulation data are shown. The X-axis denotes the number of training samples, starting from 1000 and increasing to 10,000. The Y-axis represents the total perplexity of testing samples (n = 1000). The legends of each curve are displayed above. (**e**) The distributions of topic coherence for each model are shown in mean bar plots with standard deviations. Topic coherence on each level of the scGEM is separated. *** denotes *p*-value < 0.001; ** denotes 0.001 < *p*-value < 0.01; * denotes 0.01 < *p*-value < 0.05; · denotes 0.05 < *p*-value < 0.1 and ns means non-significant. Pairwise Wilcoxon rank sum tests were used in (**a**,**d**). One-sided Wilcoxon were used in (**c**,**e**).

**Figure 5 cancers-15-04277-f005:**
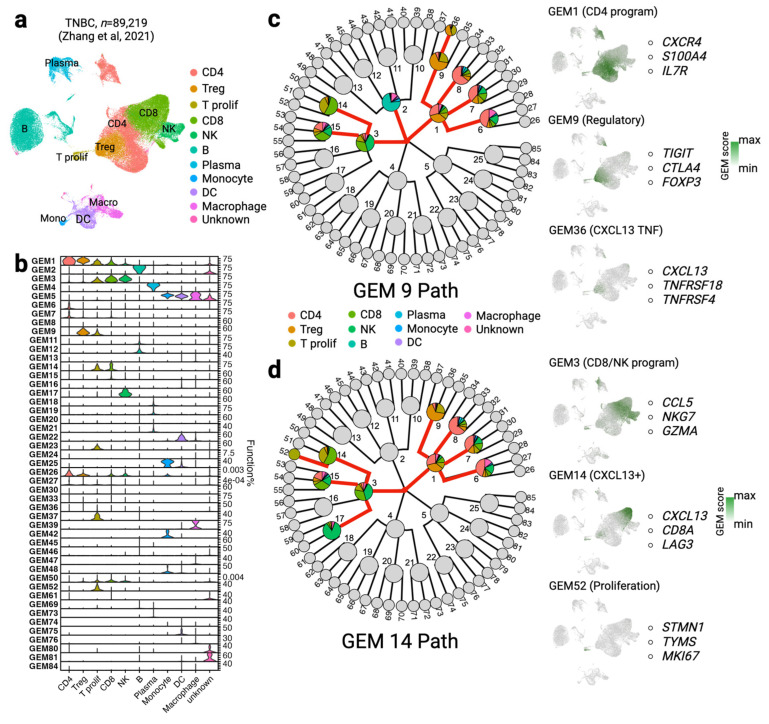
scGEM contributes to understanding the regulatory network in TNBC. (**a**) UMAP of filtered single cells of baseline TNBC tumors (n = 89,219) [32]. (**b**) GEMs that are activated in at least 1% of cells are arranged in stacked violin plots. (**c**,**d**) The searching paths of GEM 9 (**c**) and GEM 14 (**d**). Activated GEMs are colored in the pie chart, while inactivated GEMs are plotted in grey. The solid red line represents the searching path for all needed parent/sibling/child GEMs. Colors of different cell types are labeled. The UMAPs on the right exhibit the area of associated GEMs. Functional signals are revealed by the green gradient. The top three unique genes are listed to the side.

**Figure 6 cancers-15-04277-f006:**
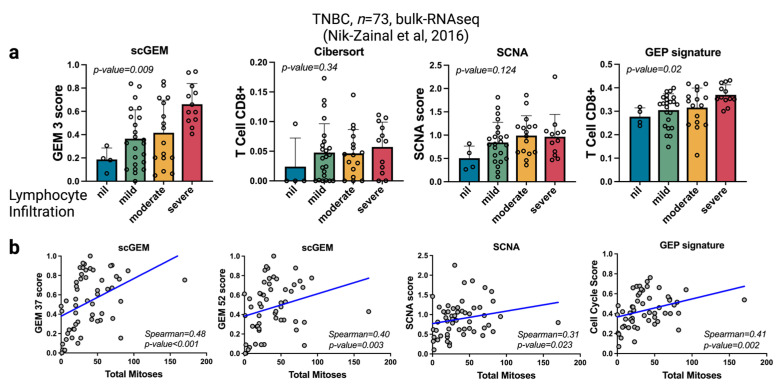
scGEM enables bulk deconvolution using knowledge from scRNAseq in TNBC. (**a**) Bar plots with standard deviation, demonstrating the distribution of T cell CD8 signals over different lymphocyte infiltration levels in the WGS560 breast cancer study [39]. T cell CD8 signals estimated by scGEM, Cibersort, SCNA score, and GEP signatures, respectively. (**b**) Scatter plots showing correlations between total mitoses and corresponding GEM scores and SCNA and GEP signatures. The Spearman correlations were measured as well.

## Data Availability

The raw data used in this study can be retrieved through the links provided in Section 2: Materials and Methods.

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
