# Peer review of "scGEM: Unveiling the Nested Tree-Structured Gene Co-Expressing Modules in Single Cell Transcriptome Data"

_cancers, 2023, doi:10.3390/cancers15174277_

Round 1

Reviewer 1 Report

The manuscript introduced scGEM, a nested nonparametric Bayesian tree-structure model, to identify cell type specific and shared GEMs in single cell transcriptome data. And the performance was compared with other methods (NFM and LDA)by doing simulations and case studies. In general, this study is well-designed and well-written but the methodology description needs to be improved.

1.     It is mentioned in Section 2.1 line 94 and 2.3.1 line 145 that UMI counts were log-normalized. Can you explain why log-normal distribution and why other probability distributions were not considered?

2.     Can you explain why Wilcox rank sum statistical test was used in the comparison of the perplexities and topic coherences between models in Section 2.5? Non-parametric methods could give lower power to identify the difference. Also, Wilcox rank sum statistical test assumes samples are independent, but they are paired in this case. Friedman test and Wilcoxon signed-rank test(pairwise comparison)might be more appropriate to detect differences in models across multiple test attempts.

3.     Please supply the description relevant to what tests do the p-values in Fig.3c and Fig.3e come from.

4.     Please describe how the GEM tree sizes of k=21, k=40 and k=85 were chosen in section 3.2. 

Author Response

Response to Reviewer 1 Comments

The manuscript introduced scGEM, a nested nonparametric Bayesian tree-structure model, to identify cell type specific and shared GEMs in single cell transcriptome data. And the performance was compared with other methods (NFM and LDA)by doing simulations and case studies. In general, this study is well-designed and well-written but the methodology description needs to be improved.

Point 1: It is mentioned in Section 2.1 line 94 and 2.3.1 line 145 that UMI counts were log-normalized. Can you explain why log-normal distribution and why other probability distributions were not considered?

Response 1: We took the log-normalization in Section 2.1 Preprocessing since it significantly reduces the mean-variance relationship in single cell transcriptome data and alleviates the skewness of the data, which are essential for downstream analysis that requires the normal distribution of the data1. Log-normal distribution was used in Section 2.3.1 Simulation in that we employed Splatter for the single cell simulation and Splatter assumes log-normal distribution to model UMI counts2.

Point 2: Can you explain why Wilcox rank sum statistical test was used in the comparison of the perplexities and topic coherences between models in Section 2.5? Non-parametric methods could give lower power to identify the difference. Also, Wilcox rank sum statistical test assumes samples are independent, but they are paired in this case. Friedman test and Wilcoxon signed-rank test(pairwise comparison)might be more appropriate to detect differences in models across multiple test attempts.

Response 2: Wilcox rank sum test was used as it relaxes the distribution of the data, which we believe a better fit to perplexity and topic coherence. We agree with you that the pairwise Wilcoxon is more appropriate in some cases. We have now clarified in Section 2.6 (original Section 2.5) that pairwise Wilcoxon was specifically used in testing perplexity as the input of data was fixed (only the model is different); thus it is reasonable to compare the results pairwisely. As for topic coherence, the contents in each topic on the same level might be different due to randomness in mini-batch initialization; besides, the number of topics on each level is dependent on the tree size. Therefore, we only use one-sided Wilcoxon for topic coherence tests. Please see highlights in revised Section 2.6 now.

Point 3: Please supply the description relevant to what tests do the p-values in Fig.3c and Fig.3e come from.

Response 3: Thank you for the comment. One-sided wilcoxon tests were used in Fig.4c and Fig.4e (original Fig.3c and Fig.3e) due to the reason in Response 2. We have now supplied the description of which tests used in Figure 4 (original Figure 3) legend and the logic in Section 2.6.

Point 4: Please describe how the GEM tree sizes of k=21, k=40 and k=85 were chosen in section 3.2. 

Response 4: Thank you for this great suggestion. We chose 5-4-3 (K=85), 4-3-2 (K=40) and 3-2-2 (K=21) at the initialization step in Section 3.3 (original Section 3.2) to assess the scGEM performance in different tree structures. As a greedy realization of Chinese restaurant process using nested K-means, we suggest the expected tree size at the initialization step to be large enough. For example, if the input single cell RNAseq data is CD45+ (T cell, B cell, NK cell, Myeloid cell, small number of other cells), like the peripheral blood mononuclear cells (PBMC) public datasets, we would suggest the number of clusters at the top level to be at least four (4-3-2). A bigger tree (5-4-3) can better represent the number of major cell types and prevent underestimation. In addition, a very large tree will result in several empty GEMs during learning. A smaller tree (3-2-2) also makes sense when the number of major cell types decreases in the training process due to the random selection of the cells. In the future,  comprehensive evaluation is required for model selection. We have now addressed this issue in Section 2.2, 3.1 and 3.3, please see corresponding highlights.

References

1              Luecken, M. D. & Theis, F. J. Current best practices in single‐cell RNA‐seq analysis: a tutorial. Molecular systems biology 15, e8746 (2019).

2              Zappia, L., Phipson, B. & Oshlack, A. Splatter: simulation of single-cell RNA sequencing data. Genome biology 18, 174 (2017).

Reviewer 2 Report

The authors developed scGEM to uncover such hidden GEMs and conducted a systematic evaluation on the model performance and comparison with existing methods. They demonstrated that scGEM has the potential to generate higher biological explanation of the GEMs using simulated and real-world datasets. There are several problems:

1.      On https://github.com/hansolo-bioinfo/scGEM, there are only several simple scripts. There are no datasets and tutorials to replicate the results.

2.      There are some similar packages, such as https://github.com/huangyuwei301/cmCluster. The authors need to make some comparisons with previous methods.

3.      The authors only listed the results. They need to add in-depth biological analysis and reveal their novel insights with their proposed method.

4.      The figures were poorly plotted. Each subfigure should be labeled and described.

5.      Have the authors tried scWGCNA? Were the gene co-expression modules (GEMs) similar?

6.      The authors should find the module-trait-relationship like WGCNA.

The authors developed scGEM to uncover such hidden GEMs and conducted a systematic evaluation on the model performance and comparison with existing methods. They demonstrated that scGEM has the potential to generate higher biological explanation of the GEMs using simulated and real-world datasets. There are several problems:

1.      On https://github.com/hansolo-bioinfo/scGEM, there are only several simple scripts. There are no datasets and tutorials to replicate the results.

2.      There are some similar packages, such as https://github.com/huangyuwei301/cmCluster. The authors need to make some comparisons with previous methods.

3.      The authors only listed the results. They need to add in-depth biological analysis and reveal their novel insights with their proposed method.

4.      The figures were poorly plotted. Each subfigure should be labeled and described.

5.      Have the authors tried scWGCNA? Were the gene co-expression modules (GEMs) similar?

6.      The authors should find the module-trait-relationship like WGCNA.

Author Response

Response to Reviewer 2 Comments

The authors developed scGEM to uncover such hidden GEMs and conducted a systematic evaluation on the model performance and comparison with existing methods. They demonstrated that scGEM has the potential to generate higher biological explanation of the GEMs using simulated and real-world datasets. There are several problems:

Point 1: On https://github.com/hansolo-bioinfo/scGEM, there are only several simple scripts. There are no datasets and tutorials to replicate the results.

Response 1: We are sorry for the inconvenience involved. We have now wrapped up the tutorial and codes into the scGEM R package. Please see the current GitHub repository.

Point 2:  There are some similar packages, such as https://github.com/huangyuwei301/cmCluster. The authors need to make some comparisons with previous methods.

Response 2: The idea of scGEM is to find gene-coexpressing module (GEM) and its quantitative distribution in each single cell. In Figure 4, we once compared the scGEM with latent Dirichlet allocation (LDA) based method and nonnegative matrix factorization (NMF) based method in terms of perplexity and topic coherence. These three methods are similar in that: they estimate the GEM and return the weight distribution of genes in each GEM; besides, they also quantify the distribution of GEM in each cell, which is of more interest to the community these years since it tells the researchers the relative proportion of cellular programs used in a certain cell; in addition, they are originated from text mining methods. From these perspectives, we kindly think that cell type clustering methods (cmCluster etc.) are not similar to our approach. To include more comparisons with previous methods, we have now evaluated the correlations of top weighted genes in scGEM and differentially expressed genes in each cell type and GEMs from scWGCNA. Please see Response 5 and new Section 3.2.

Point 3:  The authors only listed the results. They need to add in-depth biological analysis and reveal their novel insights with their proposed method.

Response 3: Thank you for the comment. To enhance clarity and enable more biological insights, we have now expanded our discussion on the functional characterization of the cell type specific GEMs and compared with typical cell cluster based method. Please see revised Section 3.1 and 3.2.

Point 4:  The figures were poorly plotted. Each subfigure should be labeled and described.

Response 4: We have thoroughly looked into each figure and now added labels in Figure 2 and Figure 5 to improve readability. The color scheme in each figure are the same. We have confirmed that each subfigure has been described in both figure legend and the main text. Of note, the overview of scGEM is described in Figure 1 legend.

Point 5:  Have the authors tried scWGCNA? Were the gene co-expression modules (GEMs) similar?

Response 5: Thank you for the suggestion. We have now compared GEMs from scGEM with scWGCNA and the highly variable marker genes in addition to the LDA and NMF based methods. We found that scGEM can represent a mixture distribution of cellular programs at a higher resolution than scWGCNA and differentially expressed genes. While one lineage of scGEM tree highly correlates with a certain cell subtype, only limited number of modules by scWGCNA associate with functional programs within the cell subtype in both peripheral blood mononuclear cell and early human brain development cells (Figure 3 and Figure S2). Please see the results in new Section 3.2.

Point 6:  The authors should find the module-trait-relationship like WGCNA.

Response 6: To determine the relationship between GEMs and trait, we used cosine similarity to measure the correlation between top weighted genes within GEM and the highly variable genes in each cell subtype. As shown in the updated Figure 3a, each cluster is associated with multiple cell type-specific GEMs, indicating a mixed distribution of cellular programs presented by scGEM modules. Please see Section 3.2.

Reviewer 3 Report

The authors have developed the scGEM method to identify gene co-expressing modules (GEMs) at the single-cell level. This method is employed to comprehend both cell-type-specific and shared GEMs within a given sample. The manuscript is well-written and intriguing. Below are my comments:

1. The authors claim that the ScGEM method can be used to understand cell differentiation processes with the assistance of GEMs. However, they haven't utilized any differentiation datasets, such as human forebrain neurogenesis, to substantiate this assertion. Thus, it would be interesting to observe the method's performance on such datasets.

2. The criteria for selecting tree-like structure levels (e.g., 5-4-3) are not clearly elucidated.

3. Figure 2a displays 85 GEM trees, but the determination of the number of GEMs and the extraction of more meaningful/significant GEMs from the others remain unclear. It would be intriguing to ascertain whether any correlations exist among GEMs.

4. Figure 2c reveals certain GEMs, like GEM 39, that are empty or low value across all cell types. The rationale for favoring GEM39 over GEM11-13 for Naïve CD4 T cells is not apparent. This information lacks clarity.

5. In Figure 2b and d, the functional characterization of GEMs should be provided.

6. Given that single-cell clusters were established using highly variable gene information, it would be valuable to explore the correlation between these genes and cell type-specific GEMs.

NA

Author Response

Response to Reviewer 3 Comments

The authors have developed the scGEM method to identify gene co-expressing modules (GEMs) at the single-cell level. This method is employed to comprehend both cell-type-specific and shared GEMs within a given sample. The manuscript is well-written and intriguing. Below are my comments:

Point 1: The authors claim that the ScGEM method can be used to understand cell differentiation processes with the assistance of GEMs. However, they haven't utilized any differentiation datasets, such as human forebrain neurogenesis, to substantiate this assertion. Thus, it would be interesting to observe the method's performance on such datasets.

Response 1: Thank you for this great suggestion. The motivation of scGEM is to identify and quantify the gene co-expressing modules (GEM) used in each cell and then determine if the modules are specific or general during cell differentiation processes. To further validate our conclusion, we have now analyzed the single-cell atlas of early human brain development1 using the same tree structure as in peripheral blood mononuclear single cell (PBMC). We also compared between the top weighted genes identified by scGEM and the marker genes within cell cluster and scWGCNA2 modules in the neuron dataset. In new Section 3.2, we demonstrated that each annotated area and Carnegie stage can be represented by a distribution of different GEMs (Figure S2). Together with our analysis in PBMC 3K dataset (new Figure 3), we showed that scGEM can capture the cellular programs at higher resolution and estimate the contributions of different GEMs in a certain cell, enabling more insights into the specialization and differentiation processes. Please see Section 3.2 now.

Point 2: The criteria for selecting tree-like structure levels (e.g., 5-4-3) are not clearly elucidated.

Response 2: In practice, it is recommended to start with a relatively large tree size to fully capture the cell programs from single cell transcriptome data, depending on the number of major cell types within. If the input has around four major cell types similar to the composition of immune cells in PBMC dataset, we would suggest a minimum of four GEMs at the top tier (4-3-2). A larger tree configuration (5-4-3) might more accurately recognize the diversity of primary cell types and avoid under-representation. We have now discussed such rationale in Section 2.2, 3.1 and 3.3. Please see associated highlights. 

Point 3: Figure 2a displays 85 GEM trees, but the determination of the number of GEMs and the extraction of more meaningful/significant GEMs from the others remain unclear. It would be intriguing to ascertain whether any correlations exist among GEMs.

Response 3: As described in Response 2, we considered that the 5-4-3 tree structure is good fit to our analysis given the number of immune cell types in PBMC and triple negative breast cancer datasets. In the first paragraph of Section 3.1, we mentioned that the GEMs are general on the top level while specialized on third level. Indeed, the meaning of GEM will become more and more specific as the topic level going deep down. The signals of cellular programs are retained on each level due to the stick-breaking process; therefore, the top level of GEM will be appropriate for common signals whereas the leaf node GEM is relevant with very unique functions. We have explained this issue in Section 2.2 now.

                  To investigate the correlation among GEMs, we have now assessed the cosine similarity of the top weighted genes between each pair of GEMs in the PBMC 3K (new Figure 3b and Figure 3c). Altogether, the meanings of GEMs correlate more on the top levels while less on the third level, implying the signals of cellular programs are less general and become more unique. Please see the revised manuscript in new Section 3.2.

Point 4: Figure 2c reveals certain GEMs, like GEM 39, that are empty or low value across all cell types. The rationale for favoring GEM39 over GEM11-13 for Naïve CD4 T cells is not apparent. This information lacks clarity.

Response 4: In Figure 2c, GEM 39 is almost invisible in the violin plot because most cells do not reveal this GEM, resulting in a distribution that only a few data points are not zero. We think this is the reason why it looks empty in the visualization and thus we showed the UMAPs as well (Figure 2b). GEM 39 is actually the representation of a very unique function in CD14+ monocytes and shared by limited number of Naïve CD4 T cells. To improve understanding, we have now provided more discussion into the functional characterization of the GEMs. Please see highlights in Section 3.1. 

Point 5: In Figure 2b and d, the functional characterization of GEMs should be provided.

Response 5: We agree with this comment. To support our conclusion and enhance readability, we have now discussed the shared and specific characterizations of myeloid GEMs in Figure 2b and T/NK GEMs in Figure 2d. Please see highlights in revised part of Section 3.1.  

Point 6: Given that single-cell clusters were established using highly variable gene information, it would be valuable to explore the correlation between these genes and cell type-specific GEMs.

Response 6: Thank you for this suggestion. To investigate the correlation between marker genes in each cell subtype and GEMs, we have now calculated the cosine similarities between the differentially expressed genes within each cluster and the Top 150 weighted genes in each GEM. As is demonstrated (new Figure 3a), each cluster correlates with several cell type-specific GEMs, suggesting a mixture distribution of cellular program represented by GEMs in cells of a certain type. In addition, we showed that the correlations among GEMs are decreasing as the tree level going down (new Figure 3b and Figure 3c), which indicates GEM are independent and specialized at lower level of the tree. Overall, we showed that GEM can represent the distribution of molecular program that a cell would employ in specialization and differentiation. Please see the description now in the new Section 3.2.

References

1              Eze, U. C., Bhaduri, A., Haeussler, M., Nowakowski, T. J. & Kriegstein, A. R. Single-cell atlas of early human brain development highlights heterogeneity of human neuroepithelial cells and early radial glia. Nat Neurosci 24, 584-594, doi:10.1038/s41593-020-00794-1 (2021).

2              Feregrino, C. & Tschopp, P. Assessing evolutionary and developmental transcriptome dynamics in homologous cell types. Dev Dyn 251, 1472-1489, doi:10.1002/dvdy.384 (2022).

Round 2

Reviewer 2 Report

The authors have answered the questions.